# PEEK–WC-Based Mixed Matrix Membranes Containing Polyimine Cages for Gas Separation

**DOI:** 10.3390/molecules26185557

**Published:** 2021-09-13

**Authors:** Marcello Monteleone, Riccardo Mobili, Chiara Milanese, Elisa Esposito, Alessio Fuoco, Sonia La Cognata, Valeria Amendola, Johannes C. Jansen

**Affiliations:** 1Institute on Membrane Technology, National Research Council of Italy (CNR-ITM), Via P. Bucci 17/C, 87036 Rende, Italy; m.monteleone@itm.cnr.it (M.M.); e.esposito@itm.cnr.it (E.E.); jc.jansen@itm.cnr.it (J.C.J.); 2Dipartimento di Chimica, Università degli Studi di Pavia, Via Taramelli 12, 27100 Pavia, Italy; riccardo.mobili01@universitadipavia.it (R.M.); chiara.milanese@unipv.it (C.M.); valeria.amendola@unipv.it (V.A.)

**Keywords:** organic cage, mixed matrix membrane, gas separation, Maxwell model, gas permeability, gas diffusion

## Abstract

Membrane-based processes are taking a more and more prominent position in the search for sustainable and energy-efficient gas separation applications. It is known that the separation performance of pure polymers may significantly be improved by the dispersion of suitable filler materials in the polymer matrix, to produce so-called mixed matrix membranes. In the present work, four different organic cages were dispersed in the poly(ether ether ketone) with cardo group, PEEK-WC. The *m*-xylyl imine and furanyl imine-based fillers yielded mechanically robust and selective films after silicone coating. Instead, poor dispersion of *p*-xylyl imine and diphenyl imine cages did not allow the formation of selective films. The H_2_, He, O_2_, N_2_, CH_4_, and CO_2_ pure gas permeability of the neat polymer and the MMMs were measured, and the effect of filler was compared with the maximum limits expected for infinitely permeable and impermeable fillers, according to the Maxwell model. Time lag measurements allowed the calculation of the diffusion coefficient and demonstrated that 20 wt % of furanyl imine cage strongly increased the diffusion coefficient of the bulkier gases and decreased the diffusion selectivity, whereas the *m*-xylyl imine cage slightly increased the diffusion coefficient and improved the size-selectivity. The performance and properties of the membranes were discussed in relation to their composition and morphology.

## 1. Introduction

Gas separation by membranes is an application in continuous growth. Because of low operating costs compared to traditional techniques, it has spread its use in several processes, including biogas [1] and natural gas separation [2], H_2_ recovery [3], post-combustion CO_2_ capture, and oxygen enriched air production [4,5]. The development of new materials that can improve separation performances represents a continuous request for the research field. Nowadays, the availability of materials, which are above the upper bound in the Robeson plot and combine a high permeability with a high selectivity, is very low. In general, polymers with high permeability (usually rubbers or high free volume polymers) show a low selectivity, while highly selective materials (usually glassy polymers) are generally characterized by a low permeability. The addition of fillers to a polymer is a route for the preparation of mixed matrix membranes (MMMs), in which the final performances are obtained by exploiting the synergic action of the easy processability of polymers with the high gas separation performance of the filler. Several additives such as zeolites [6], porous metal-organic frameworks (MOFs) [7,8,9], and organic cages [10,11] were used for this purpose. However, compatibility between the polymer and the filler is essential for the formation of defect-free membranes. Agglomeration and sedimentation of particles, as well as the bad polymer/filler adhesion, are problems which can often occur, inducing the formation of defects in the membranes [9].

In this work, we study the effect of four different polyimine cages (*m*-xylyl, *p*-xylyl, diphenyl and furanyl-based systems) on the gas transport properties of a poly(ether ether ketone) with cardo group (PEEK-WC, Figure 1). The latter is a glassy polymer that is soluble in several organic solvents due to the presence of the Cardo-group in the backbone, unlike the classical poly (ether ether ketone) (PEEK) and poly (ether ketone) (PEK) [12], and its good solubility in low boiling solvents like chloroform is useful for the preparation of dense gas separation membranes by solvent evaporation. Recently, PEEK-WC was loaded with MOF, improving the transport properties in gas separation [13].

Cage-like organic molecules [14,15], cryptands and cryptates in particular [16], are molecular systems with tridimensional cavities that are well known in the literature for many applications in solution, including host–guest chemistry, catalysis, and drug delivery [17,18,19]. After the pioneering work by Cooper et al. [20], porous organic cages have also raised attention in solid state as molecular materials for gas separation processes [21,22,23]. The cages proposed for these applications are generally characterized by a large tridimensional cavity, and their porosity is permanent in the solid state, thanks to the rigid molecular structure. Compared to other porous materials employed in gas separation (such as zeolites, MOFs, covalent organic frameworks) [24], organic cages are generally more soluble in organic solvents [25], which makes them more attractive during the preparation of MMMs, since they can be more easily dispersed in the casting solution [10,11,26]. Among organic cages, bistren azacryptands have been extensively described in the literature and they are easily obtained in high yields, starting from a flexible commercial polyamine (i.e., tris(2-ethylamino)amine, *tren*) and a properly chosen dialdehyde [27,28]. The rigidity and size of the polyimine cage framework can be modulated by changing the spacers connecting the *tren* subunits. These features, which are also maintained after reduction to the amino form, are employed by several groups to enhance the cage selectivity towards guest molecules in solution. A number of bistren azacryptands (in particular polyimine systems) has also been reported as solid materials for the electrochemical reduction of CO_2_ to CO and for CO_2_ sequestration processes from multicomponent gas streams [29,30,31]. However, as far as we know, this particular class of molecules has never been tested in MMMs. This work is aimed to fill this gap by developing a series of MMMs, with improved performance in terms of gas permeability and selectivity, containing polyimine bistren cryptands as fillers. To achieve this goal, we considered four cages featuring different spacers between the *tren* subunits (Figure 1). The choice of spacers was based on their length (e.g., 1,3-xylyl vs. 1,4-xylyl; 4,4′-diphenyl vs. 1,4-xylyl), rigidity and presence of heteroatoms (xylyl vs. furanyl).

The most commonly used model to describe the gas transport in polymeric membranes is the solution–diffusion model. According to this model, the permeation occurs in three steps. In the first step, the penetrating gas molecules dissolve in the membrane on the feed side. In the second step, they diffuse through polymeric matrix from the feed side to the opposite side (permeate). In the third and final step, the gas molecules desorb from the permeate side of the membrane into the permeate reservoir. The diffusion process, which is the rate-determining step and is slower than the other two, depends on the size of the penetrating gas and on the space available between the polymer chains (free volume) and on their rigidity. According to the solution–diffusion model, the permeability (*P*) of a gas through a membrane is expressed as the product between the diffusivity (*D*) and solubility (*S*):*P* = *D* × *S*(1)

Permeability is a fundamental parameter that expresses the ability of the membrane material to be permeated by a specific gas. This value represents the flux of permeate generated by a pressure difference between the two sides of the membrane, normalized for the surface area and the thickness of the membrane, and it is commonly given in the unit Barrer (1 Barrer = 10^−10^ cm^3^_STP_ cm cm^−2^ s^−1^ cmHg^−1^).

For a pair of gases *a* and *b*, the selectivity (*α_a,b_*) is determined by the ratio of the permeability of gas *a* and the permeability of gas *b*:(2)$$\alpha_{a,b\;}=\frac{P_{a}}{P_{b}\;}$$

In mixed matrix membranes, the presence of a filler influences the transport properties with respect to that of the pure polymer. Several models were proposed to describe this effect. One of the simplest and most applied model is the Maxwell model [32,33], valid when spherical fillers are present in the membrane at low concentration, up to approximately 30 vol %. The permeability of a MMM, *P_MMM_*, is given by the following equation:(3)$$P_{MMM}=P_{C}{\left[\frac{P_{d}+2P_{c}-2\Phi_{d\;}\left(P_{c}-P_{d}\right)}{P_{d}+2P_{c}+\Phi_{d\;}\left(P_{c}-P_{d}\right)}\right]}_{\;}$$
where *P_c_* and *P_d_* are the permeability of the continuous and dispersed phase, respectively, while Φ*_d_* is the volume fraction of the dispersed phase. Thus, if the dispersed phase is more permeable than the continuous phase, the overall permeability increases, with a maximum theoretical limit given by *P_d_* = ∞:(4)$$P_{MMM,max}=P_{C}{\left[\frac{1+2\Phi_{d\;}}{1-\Phi_{d\;}}\right]}_{\;}$$

The overall permeability decreases if the permeability of the dispersed phase is lower than that of the bulk polymer, with a minimum limit for *P_d_* = 0:(5)$$P_{MMM,min}=P_{C}{\left[\frac{1-\Phi_{d\;}}{1+0.5\Phi_{d\;}}\right]}_{\;}$$

More complex situations occur for non-spherical particles, or when the polymer/filler interaction affects the properties of the bulk and creates a stiffer interface or an interface with higher free volume.

## 2. Results and Discussion

### 2.1. Cages Preparation and Characterization

The polyimine bistren cages **1**–**4** (Figure 1) were obtained in high yields (>70%) following the procedures described in the literature [34], which consist in the condensation between *tren* and the chosen dialdehyde, mixed in a 2:3 molar ratio in acetonitrile. The pure products, precipitated from the reaction mixture, were isolated by filtration and dried under vacuum at room temperature for 48 h. Fura precipitated as brownish microcrystals from the reaction mixture, while *m*-xy formed agglomerates of a white microcrystalline powder. On the other hand, *p*-xy and diphen precipitated as colorless solids, which tended to compact under drying conditions. The ^1^H-NMR spectra of cages **1**–**4** (Appendix A) in CDCl_3_ correspond to those reported in the literature [34]; FTIR-ATR spectra further confirm the presence of imine bonds (Appendix A) as it results from the typical peaks (between 1629 and 1641 cm^−1^) due to the stretching of the C=N bond. The solubility of the compounds was determined experimentally at room temperature in various solvents. The results are listed in Table 1. These tests showed that Fura and *m*-xy cages are generally more soluble than *p*-xy and diphen in common solvents used in membrane preparation.

Fura and *m*-xy were found to be the most crystalline samples among the investigated systems, as indicated by the XRPD patterns (Figure 2a,b). Instead, an amorphous halo is evident in both *p*-xy and the diphenyl cage (Figure 2c,d), where few peaks are superimposed on a very broad signal in the 10–30° angular range. Notably, the experimental patterns of Fura and *m*-xy cages are in very good agreement with the patterns predicted by the Mercury software [35] on the basis of single crystal data reported in the literature [36,37]. In the rhombohedral crystals of the *m*-xy cage, reported by Nelson et al. [37], no evidence of intermolecular interactions or of H-bonded solvent molecules was found. On the other hand, the orthorhombic Fura crystals [36] presented one crystallization water molecule per cage, H-bonded to the imine bonds of two adjacent cryptand molecules.

TGA (Figure 3) and DSC (Appendix A) measurements reveal that the *m*-xy and diphenyl cages are the thermally most stable species among the investigated samples (Figure 3b,d, Appendix A). Apart from a slow step of mass loss, attributable to the release of the crystal water in Fura or of adsorbed solvent in all cages, these samples started to decompose at temperatures of 200 °C and higher. The confirmation of the irreversible nature of these processes, due to the cage decomposition, is given by the cooling curves in the DSC profiles that do not show any relevant signal in the considered temperature range. The TGA curve of *p*-xy (Figure 3c) shows two small mass losses between room temperature and 100 °C, attributable to solvent release, while decomposition starts at about 180 °C. In the case of Fura, beside the solvent release, decomposition starts at about 200 °C, as evident by the derivative of the TGA curve (DTG, dm/dT trace in Figure 3a) and by the events in the DSC profile (Appendix A).

The morphological analysis of the cages by SEM shows that most of the powders have an irregular shape and heterogeneous particles size (Figure 4). In the case of Fura, the SEM images displayed crystals of various size, whose shape is consistent with an orthorhombic lattice, that could be hypothesized after comparing the experimental XRPD pattern with that predicted on the basis of published single crystal analysis (Figure 2a) [36]. The morphological analysis of *m*-xy showed large and irregular aggregates; while for *p*-xy, fine spherical-shaped particles could be seen even at the highest magnification. The SEM image of the diphen cage displayed heterogeneous agglomerates of flakes and needle-like crystals.

### 2.2. Membrane Preparation and Characterization

The solubility of the cages is not high enough to make a concentrated solution in chloroform (Table 1); therefore, the dispersion of the cages in the polymer solution was chosen as the route for membrane preparation in this work. Moreover, only two cages were suitable for MMMs preparation with the used protocol and polymer, the *m*-xy and Fura. Thus, successfully prepared MMMs PEEK-WC/*m*-xy and PEEK-WC/Fura did not show any macroscopic defects or inhomogeneities visible to the naked eye, suggesting an even dispersion of the fillers. They were only slightly opaque due to the different refractive index of the polymer and the filler materials and had a thickness of 42 μm and 78 μm for PEEK-WC/*m*-xy and PEEK-WC/Fura, respectively. The membranes were robust and could be handled without particular attention. Instead, the other two cages, *p*-xy and diphen, were not suitable for the preparation of homogeneous and stable membranes due to their poor dispersion in the polymeric solution. This is confirmed by the SEM images, showing evident clustering of the *p*-xy cage that hinders its homogenous dispersion (Figure 4).

The MMMs’ morphology was studied by SEM analysis of the top and bottom surfaces, as well as their cross-section (Figure 5). 

It should be noted that the fully organic nature of the polymer and the cages does not give a strong contrast in the SEM pictures, and thus the cages are not well visible. The PEEK-WC/*m*-xy membrane cross-section shows a well-defined dense layer and both surfaces show no evident pores or defects, confirming the good dispersion of the cage in the membrane. Instead, the PEEK-WC/Fura membrane has irregularities on both surfaces, such as pores and cracks, and its cross-section is not uniform. The presence of these defects is a symptom of a membrane with low selectivity that needs to be coated to perform defect healing. Herein, we have coated the membranes with a dense PDMS layer, which is a widely used procedure for the correction of pinhole defects in membranes for gas separation [38,39,40,41]. The effect of PDMS on the diffusive transport through the dense membrane is negligible when there is a difference of several orders of magnitude in their intrinsic permeability coefficients, as is the case for our PEEK–WC-based MMMs. Instead, PDMS completely blocks the Knudsen diffusion through the pinhole defects (Figure 6), and the transport through the coated membrane is governed by the solution–diffusion mechanism.

### 2.3. Pure Gas Transport Properties

An example of two permeation curves, determined by the so-called time lag method in a fixed-volume pressure-increase setup, and described by (6), is given in Figure 6. The permeate pressure of CO_2_ in the PEEK-WC/*m*-xy membrane is plotted as a function of time, before and after defect healing with PDMS. From the immediate pressure increase and the very steep slope of the uncoated sample, it is evident that the PDMS coating is needed to correct pinhole defects and to obtain a curve where the determination of the time lag is clear and well defined. This allows the determination of the gas transport parameters of the MMM, i.e., the permeability and diffusion coefficients of the gases and, indirectly, the solubility. The resulting flat baseline in the PDMS-coated membrane, i.e., the tangent to the very initial part of the curve defined by the term (*dp/dt*)_0_ in (Equations (6) and (7)), confirms that leak flow through remaining pinhole defects is negligible for CO_2_. Wherever this is not the case, a baseline correction was applied via (6) and (7). As described previously, this procedure allows the correct calculation of the values of *P*, *D,* and *S* of membranes with few defects [42].

The results of the permeation tests with six pure gases at 25 °C are collected in Table 2. The measurements were performed in the order H_2_, He, O_2_, N_2_, CH_4_, and finally CO_2_. Tests with O_2_, N_2_ were repeated at the end of the cycle in order to confirm that there is no change in the material due to physical aging or plasticization by CO_2_. The incorporation of the cages induced different effects in the transport properties compared to the neat polymer, depending on the cage type. A decrease in permeability for all gases was observed in the presence of the *m*-xy cage, while a gain in selectivity was found for the gas pairs CO_2_/CH_4_, CO_2_/N_2_, O_2_/N_2_, and He/CH_4_, as displayed in the Robeson diagrams in Figure 7. The remarkable gain in selectivity for the O_2_/N_2_ gas pair leads to an overall improvement of the membrane separation performance that approaches the 1991 upper bound (Figure 7c). The opposite behavior occurs with the Fura cage, where the permeability of all gases increases, while the selectivity decreases for most gas pairs, except for CO_2_/N_2_ that remains almost constant. Especially with *m*-xy, the herein proposed MMM is among the best performing membranes in the class of poly(aryl ether) and poly(aryl ether ketone)-based systems in terms of selectivity, while with Fura, they are among the best in terms of permeability.

On the basis of the Maxwell Equation (3), the decrease in permeability in the presence of *m*-xy, and the increase in permeability for Fura suggests that these fillers have a lower and higher permeability than the polymer, respectively. Assuming a density of approx. 0.5 g cm^−3^ for the cages [29] and 1.249 g cm^−3^ for the PEEK-WC [43], the volume fraction of the cages is 38.4% in the MMMs with 20 wt % of cages, and (4) and (5) predict that their permeability falls in the interval 0.517 *P*_c_ < *P_MMM_* < 2.87 *P*_c_. In practice, the decrease for many gases in the *m*-xy-based MMM is close to or even slightly larger than predicted by (5), suggesting that we deal with almost impermeable fillers or that the fillers also affect the bulk properties of the polymer. Fortunately, along with a decrease in permeability, we see an increase in selectivity, indicating that the separation performance of the membrane increases, in spite of its lower productivity. Instead, the increase in permeability with Fura is much higher than predicted by the Maxwell equation and at the same time there is a strong reduction in diffusion selectivity with this cage (Figure 8 and discussion below). This is probably related to the presence of additional free volume and might indicate the formation of nonselective diffusion paths around the cages due to poor adhesion, or it indicates the presence of voids between poorly dispersed clusters.

**Figure 7 molecules-26-05557-f007:**
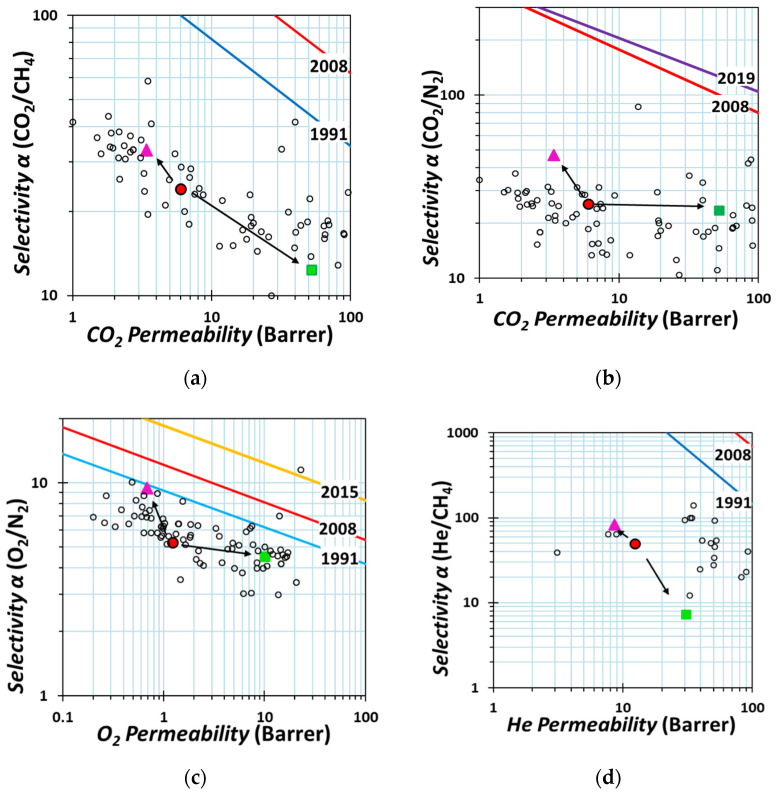
Robeson diagrams for the (**a**) CO_2_/CH_4_, (**b**) CO_2_/N_2_, (**c**) O_2_/N_2_, and (**d**) He/CH_4_ gas pairs with the upper bounds represented by blue lines for 1991 [44], red lines for 2008 [45], yellow lines for 2015 (O_2_/N_2_) [46], and purple lines for 2019(CO_2_/N_2_) [47]. The gas permeability data for PEEK-WC are reported with red circles ●, with pink triangles for PEEK–WC/*m*-xy ▲, and with green squares for PEEK–WC/Fura ■. Empty symbols are literature data from the database of the Membrane Society of Australasia (MSA) for Poly(aryl ethers) and Poly(aryl ether ketones)-based membranes (https://membrane-australasia.org/msa-activities/polymer-gas-separation-membrane-database/, last accessed on 29 August 2021).

A decrease of the solubility coefficient for nearly all bulkier gases was observed in both mixed matrix membranes. With the *m*-xy cage, the permeability also decreases, since the loss in solubility is not balanced by an increase in diffusivity. Instead, with the Fura cage, the decrease in solubility is overcompensated by a dramatic increase in the gas diffusion coefficients, leading to an overall increase in permeability.

The logarithm of the diffusion coefficient shows a linear correlation with the square of the gas diameter, *d*^2^*_eff_*, for all investigated membranes (Figure 8). This trend indicates that the gas transport in all membranes follows the solution–diffusion model and no anomalies are taking place [48]. However, while the PEEK–WC/*m*-xy MMM has a similar behaviour with respect to the neat PEEK–WC membrane, the PEEK–WC/Fura has a much gentler slope, indicating weaker size-sieving behaviour. This is in agreement with the hypothesis, formulated during the Maxwell analysis, on the formation of nonselective free volume elements, for instance in the form of diffusion paths around the cages due to poor adhesion or due to voids between poorly dispersed clusters.

It should be noted that only in the neat PEEK–WC membrane, the diffusion coefficient of CO_2_ is higher than that of N_2_, as expected on the basis of the effective gas diameters, while in both MMMs, the CO_2_ diffusion coefficient does not increase as much as that of N_2_ and the order of the two gases is inverted. An unexpectedly slow transient is usually a sign of specific interaction of CO_2_ with the cage, or simply a higher solubility in internal voids of the dispersed phase. Further control of the cage size down to nanometer-scale is needed to be able to produce successfully integrally skinned or thin film composite PEEK-WC membranes that may have a selective layer below 100 nm thick [49].

**Figure 8 molecules-26-05557-f008:**
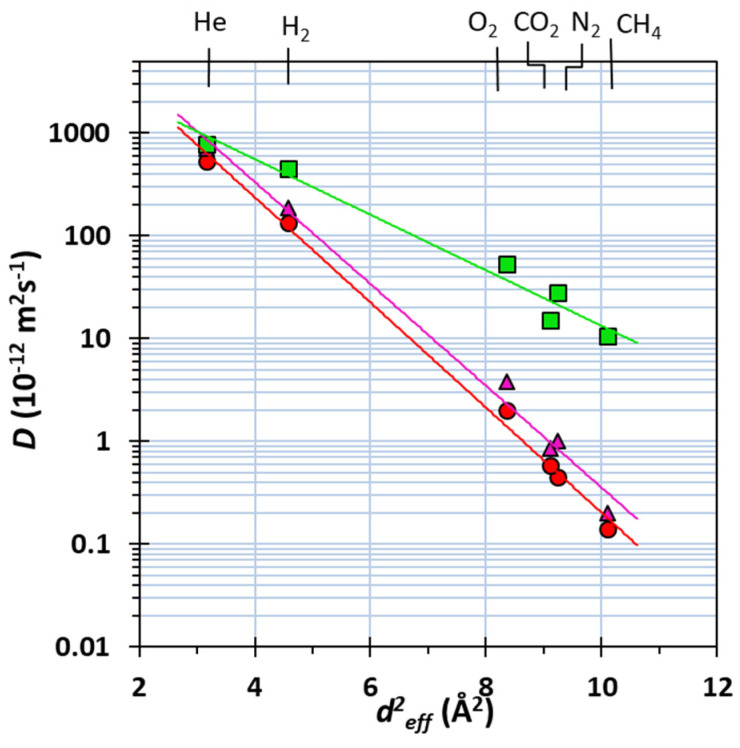
Correlation of the effective diffusion coefficient of six light gases as a function of their molecular diameter, as defined by Teplyakov and Meares [50], for the PEEK–WC ●, PEEK–WC/*m*-xy ▲, and PEEK–WC/Fura ■ membranes.

## 3. Materials and Methods

### 3.1. Materials for Synthesis and Characterization

Solvents and chemicals used for syntheses were HPLC grade. Acetonitrile, diethyl ether, tris(2-ethylamino) amine 96%, 2,5-furandicarbaldehyde 97%, isophthalaldehyde 97%, terephthalaldehyde reagentPlus 99% and deuterated solvents used for NMR analysis (CDCl_3_) were purchased from Sigma-Aldrich, Merck Italia (Milano, Italy). Diphenyl-4,4′-dicarboxyaldehyde was synthesized based on a procedure already described by our group [51]. PEEK-WC was supplied by the Institute of Applied Chemistry, Changchun, China. PDMS resin ELASTOSIL^®^ M 4601 A/B was provided by Wacker Chemie AG. (Munich, Germany). Chloroform AnalaR NORMAPUR^®^, was supplied by VWR International srl, Milano, Italy. Pure gases H_2_, He N_2_, O_2_, CH_4_, CO_2_ (99.99+%) used for permeation tests were purchased from SAPIO, Monza, Italy.

### 3.2. Syntheses and Characterization of Cages **1**–**4**

Syntheses and characterizations of the investigated cages (see **1**–**4** in Figure 1) have been already reported [34,36,37,52]. For this work, Fura, *m*-xy, *p*-xy, and diphen were obtained following the procedure recently described by Lehn et al. [34], using acetonitrile as the solvent for the synthesis. The cages were precipitated from the reaction mixtures, collected by filtration, and dried under vacuum. The obtained polyimine cages were employed in the preparation of MMMs without further purification.

#### Cages Characterization

The solubility of cages **1**–**4** was tested in various solvents (MeOH, EtOH, CHCl_3_, CH_2_Cl_2_, and THF at T = 25 °C): about 2 mg of each cage was weighted in a test tube and solvent was added in small portions until complete dissolution of the powder at room temperature (upon sonication). The experiment was then repeated for each solvent. The obtained results are shown in Table 1.

^1^H-NMR spectra were recorded on a Bruker ADVANCE 400 spectrometer (operating at 9.37 T, 400 MHz). Chemical shifts are reported in ppm with the residual solvent as internal reference. NMR spectra were recorded at 25.0 °C.

X-ray powder diffraction (XRPD) measurements were performed at room temperature on the powders of the four cage samples after manual grinding in an agate mortar using a Bruker D5005 diffractometer (Bruker Corporation, Billerica, MA, USA) with CuKa radiation, graphite monochromator, and scintillation detector. The measurements were performed from 3° to 70° with step scan mode: scan step 0.02°, counting time 10 s per step; X-ray tube working conditions: 40 kV and 40 mA.

For the Fourier Trasform Infrared analysis (FTIR), a Nicolet FTIR iS10 spectrometer (Nicolet, Madison, WI, USA) equipped with attenuated total reflectance (ATR) sampling accessory (Smart iTR with diamond plate) was used. Thirty-two scans in the 4000–600 cm^−1^ range at 4 cm^−1^ resolution were coadded. Well-ground powder samples were used and spectra were obtained after pressing the sample onto the ATR diamond crystal at room temperature (20 °C). Peak wavenumbers were attributed by using the “Find peaks” function of the OMNIC™ Spectra Software.

Thermogravimetric analysis (TGA) was performed by a Q5000 apparatus (TA Instruments, New Castle, DE, USA) interfaced with a TA5000 data station under nitrogen flux (10 mL min^−1^) in a platinum pan by heating about 3 mg of sample from room temperature up to 500 °C (heating rate 5 K min^−1^).

Differential scanning calorimetry (DSC) was performed by a Q2000 apparatus (TA Instruments, New Castle, DE, USA) interfaced with a TA5000 data station by heating about 3 mg of powder in an open aluminum crucible from −50 °C to 350 °C (heating rate 5 K min^−1^) under nitrogen flux (50 mL min^−1^). Three independent measurements on three different samples were performed for each cage. The temperature accuracy of the instrument is ±0.1 °C, the precision is ±0.01 °C, and the calorimetric reproducibility is ±0.05%. TGA and DSC data were analyzed by the Universal Analysis software by TA Instruments.

Scanning electron microscopy (SEM) analysis of the powder was performed by Phenom Pro X desktop SEM, Phenom-World. The images were acquired with an accelerating voltage of 10 kV at different magnification: 1000×, 5000×, and 20,000×.

### 3.3. Membranes Preparation

Mixed matrix membranes in PEEK–WC were prepared with a loading of 20 wt % of each cage on the basis of polymer mass. PEEK–WC was dissolved in chloroform at 3 wt % under magnetic stirring for 24 h at room temperature. Then, the obtained solution was filtered by glass syringe filter of 3.1 µm. The cages were dispersed in chloroform and sonicated for 30 min before their addition to the PEEK–WC solution. The PEEK–WC/Cage suspension was sonicated for 5 h at room temperature in order to obtain a homogeneous dispersion. The solutions were casted in a Teflon petri dish and the self-standing dense membranes were obtained by solvent evaporation at 35 °C. Finally, the obtained membranes were coated with PDMS Elastosil M 4601 to perform the defect healing process. The two-component PDMS, A and B, were mixed in weight ratio 9:1 according to the instructions of the supplier without the use of a solvent. A film of ca. 25 µm of the resin was applied on the surface of the membrane by a casting knife. The coated membranes were kept at room temperature to complete the crosslinking in 24 h.

### 3.4. Membranes Characterization

#### 3.4.1. Morphological Characterization Membranes (SEM)

Scanning electron microscopy (SEM) analysis of the membranes was performed by a Phenom Pro X desktop SEM, Phenom-World. The images were acquired with an accelerating voltage of 15 kV at different magnifications: 1000×, 5000×, and 20,000×.

#### 3.4.2. Single Gas Permeation Method

Single gas permeation measurements were performed on circular membranes (exposed area 13.84 cm^2^) at 25 °C and at a feed pressure of 1 bar by a fixed volume/pressure increase instrument designed by HZG and constructed by EESR (Geesthacht, Germany). Further details on the measurement protocols and data treatment are described in a previous paper [53]. Before each measurement, the membranes were evacuated in the testing cell by a turbo-molecular pump until complete desorption of all previously adsorbed gases and humidity. For the same reason, between two consecutive tests, the membranes were evacuated for a time equal to 10 times the time lag of the previous gas.

The time lag method was used for the determination of the permeability (*P*), diffusion (*D*), and solubility coefficients (*S*), which can be obtained from the increase of the permeate pressure, *p_t_*, as a function of time, t, after exposure of the membrane to the gas [54]:(6)$$\begin{array}{ll} p_{t} & =p_{0}+{\left(\frac{dp}{dt}\right)}_{0}\cdot t+\frac{RT}{V_{P}\cdot V_{m}}\cdot A\cdot l\cdot p_{f}\cdot S\\ & \;\times (\frac{D\cdot t}{l^{2}}-\frac{1}{6}-\frac{2}{\pi^{2}}\overset{\infty }{\underset{n=1}{\sum }}\frac{{\left(-1\right)}^{n}}{n^{2}}\exp\left(-\frac{D\cdot n^{2}\cdot \pi^{2}\cdot t}{l^{2}}\right)) \end{array}$$
where *p*_0_ and (*dp/dt*)_0_ are the starting pressure and baseline slope, respectively, which should be negligible in a well-evacuated and leak free membrane and permeability instrument. *R* is the universal gas constant, *T* the absolute temperature, *V_P_* the permeate volume, *V_m_* the molar volume of a gas at standard temperature and pressure [22.41 × 10^−3^ m^3^_STP_ mol^−1^ at 0 °C and 1 atm], *A* the exposed membrane area, *l* its thickness, *p_f_* the feed pressure, *S* the gas solubility and *D* the diffusion coefficient. The permeability *P* was obtained from the permeation curve (7) in the pseudo steady-state:(7)$$p_{t}=p_{0}+{\left(\frac{dp}{dt}\right)}_{0\;}.t+\frac{RT\;A}{V_{p}\;V_{m}\;}.\;\frac{p_{f}\;P}{l}\;{\left(t-\frac{l^{2}}{6D}\right)}_{\;}$$

The diffusion coefficient is inversely proportional to time lag (Θ) and was calculated from (8):(8)$$\Theta =\frac{l^{2}}{6D}$$

The solubility coefficient (*S*), was calculated from the solution–diffusion transport model (9):*S* = *P*/*D*(9)

## 4. Conclusions

In this work, four different polyimine bistren cages were studied as fillers for the preparation of PEEK–WC-based mixed matrix membranes for gas separation. Only the two cages *m*-xy and Fura proved to be suitable for obtaining robust dense MMMs with few enough pinhole defects to be healed by PDMS coating. Instead, inhomogeneous dispersions were obtained with the two cages diphen and *p*-xy in the polymer solution, which led to non-uniform and highly defective membranes also after the PDMS coating.

Compared to the pure polymer, the permeability of all the tested gases increased in presence of Fura, and decreased with the *m*-xy cage. In terms of selectivity, the presence of *m*-xy increases the selectivity for the gas pairs CO_2_/CH_4_, CO_2_/N_2_, O_2_/N_2_, and He/CH_4_, whereas the CO_2_/N_2_ selectivity did not significantly change with Fura and it decreased for the other gas pairs. The behavior can be explained with the Maxwell model, which indicates that the permeability of *m*-xy is lower than that of the polymer matrix whereas that of Fura is much higher. The exceptionally high permeability in the presence of Fura suggests that there are other permeation pathways, probably in the cage/polymer interface.

The use of these cages as fillers in the polymer matrix generally increases the diffusivity for all the investigated gases. In particular, the slight increase in diffusivity promoted by *m*-xy enhances the size-selectivity of the membrane. The diffusivity of the bulkiest gas molecules increases remarkably with the Fura cage, resulting in an evident loss in diffusion selectivity. The gas transport follows the solution–diffusion mechanism also after the dispersion of the cages in the membranes, because it does not change the linear correlation between the logarithm of the diffusion coefficient and the square of the effective gas diameter *d^2^_eff_*. The obtained results are a good starting point for studying the effect of cages concentration on the gas transport properties of MMMs in different polymer materials, which will be evaluated in detail in further studies. Since integrally skinned or thin film composite PEEK–WC membranes may have a selective layer thickness below 100 nm [49], future work should also focus on further control of the cage size down to nanometer-scale, in order to produce realistic membranes successfully.

## Figures and Tables

**Figure 1 molecules-26-05557-f001:**
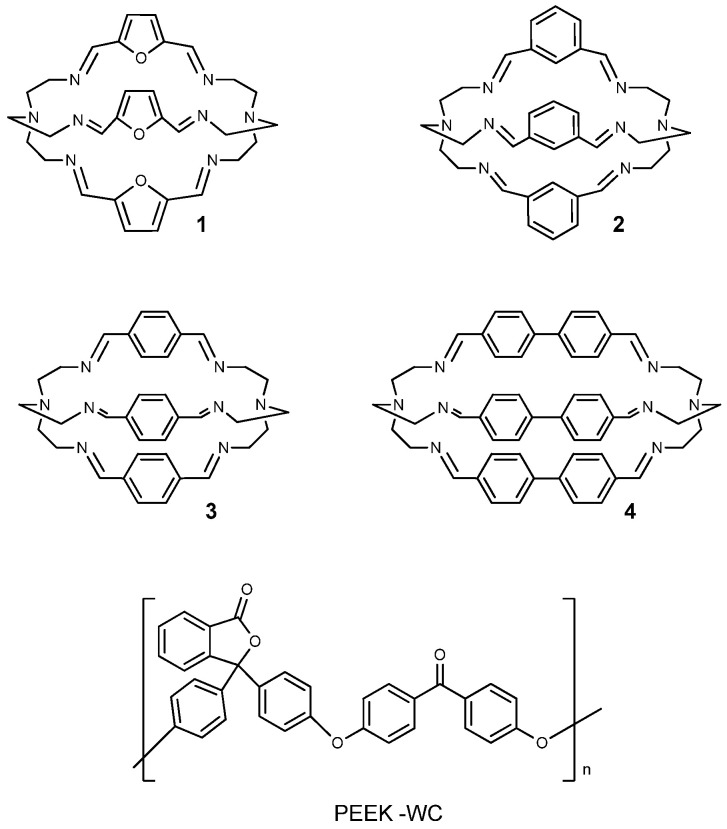
Structure of the cages (**1** = furanyl, **2** = *m*-xylyl, **3** = *p*-xylyl, **4** = diphenyl) and the polymer PEEK-WC used in the present work.

**Figure 2 molecules-26-05557-f002:**
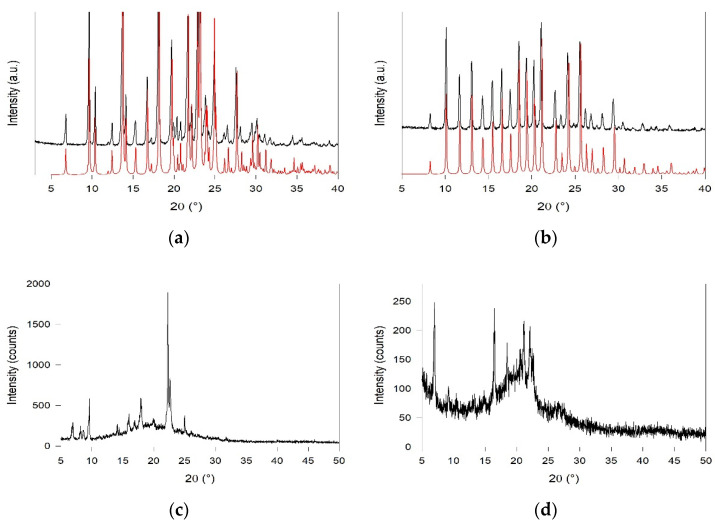
XRPD patterns of (**a**) Fura and (**b**) *m*-xy, (**c**) *p*-xy and (**d**) diphen samples. Black lines: experimental patterns, and red line: patterns predicted by the Mercury software [35] on the basis of single crystal data reported in the literature [36,37].

**Figure 3 molecules-26-05557-f003:**
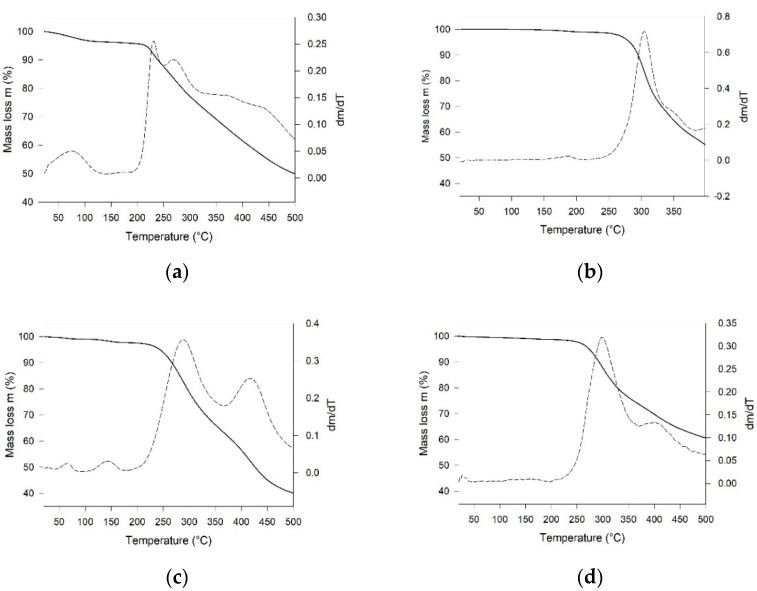
Thermogravimetric (solid line) and DTG curves (dashed line) for (**a**) Fura, (**b**) *m*-xy, (**c**) *p*-xy, and (**d**) diphen cages.

**Figure 4 molecules-26-05557-f004:**
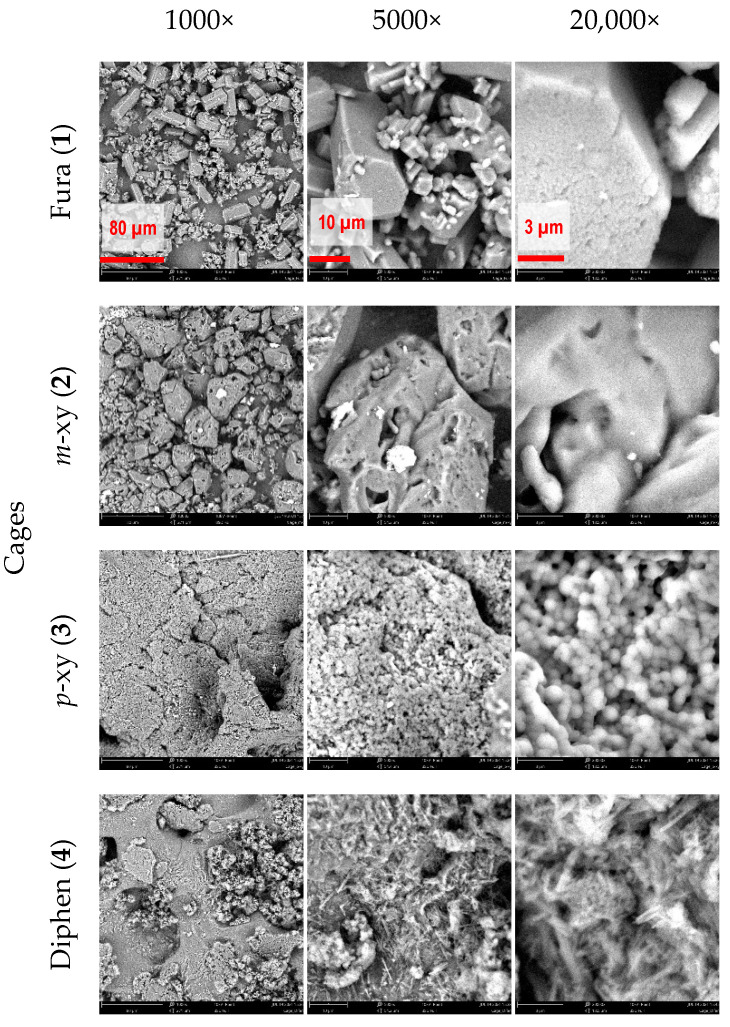
SEM images of Cages **1**–**4** as collected in their powder form at different magnifications.

**Figure 5 molecules-26-05557-f005:**
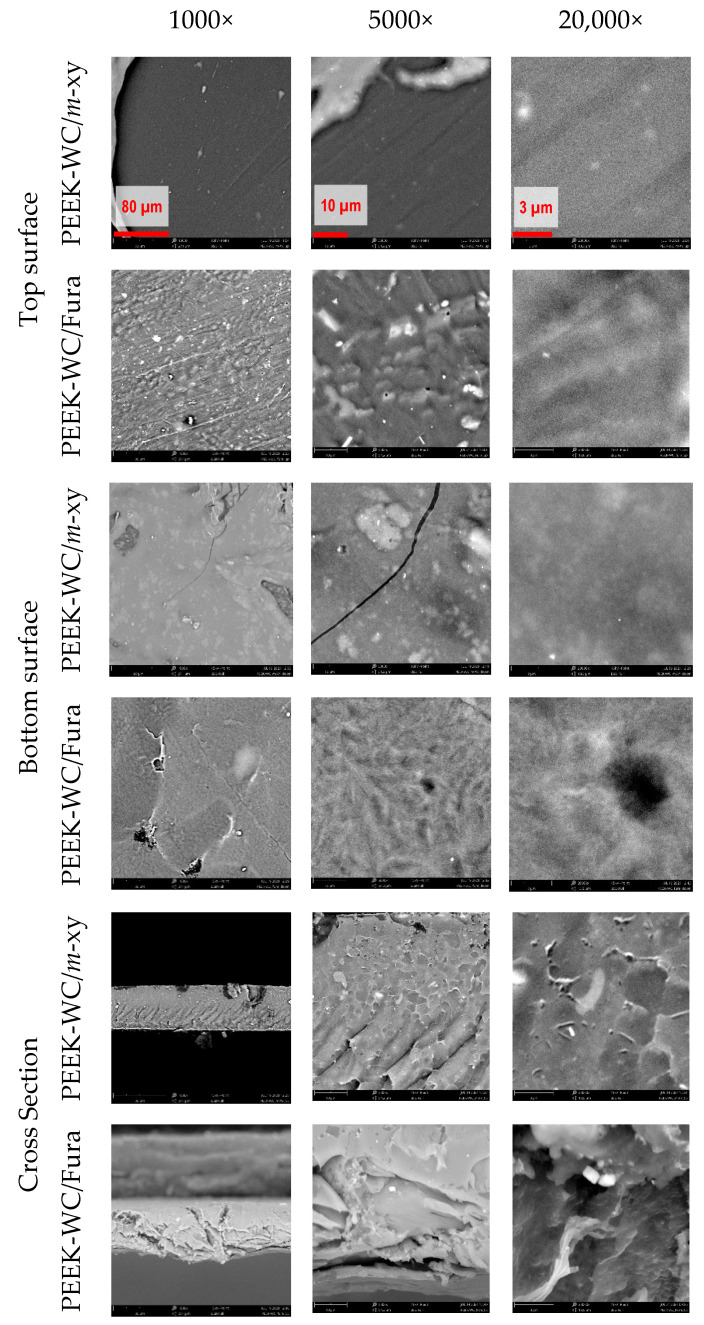
SEM images of the PEEK–WC-based membranes and the cages Fura (**1**) and *m*-xy (**2**) at different magnifications.

**Figure 6 molecules-26-05557-f006:**
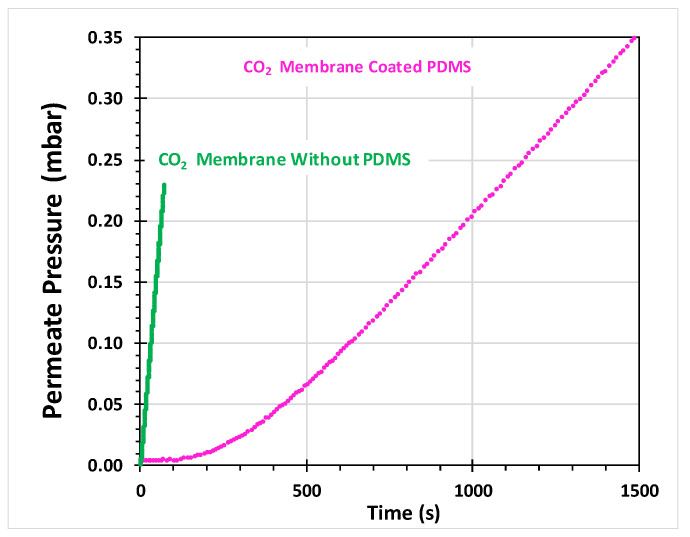
Example of CO_2_ pure gas permeation curves at 25 °C and 1 bar feed pressure of membrane PEEK-WC/*m*-xy with and without silicone coating, showing the effective healing of pinhole defects by the flat baseline (dashed line).

**Table 1 molecules-26-05557-t001:** Solubilities of the prepared cages.

			Solubility (wt/vol %) in the Given Solvents				
No.	Code	Polyimine Cages	MeOH	EtOH	CHCl_3_	CH_2_Cl_2_	THF
1	Fura	furanyl-based	0.10	0.08	0.13	0.10	X
2	*m*-xy	*m*-xylyl-based	0.13	0.10	0.40	0.40	0.20
3	*p*-xy	*p*-xylyl-based	X	X	0.20	0.13	X
4	diphen	diphenyl-based	X	X	0.10	0.07	X

X = completely insoluble in the chosen solvent.

**Table 2 molecules-26-05557-t002:** Pure gas permeability, solubility, and diffusion coefficients and relative selectivity for the neat PEEK–WC, PEEK–WC/*m*-xy, and PEEK–WC/Fura membranes.

**Membrane**	**Permeability [Barrer]**						** *α* ** **(*P_a_*/*P_b_*)**			
	**N_2_**	**O_2_**	**CO_2_**	**CH_4_**	**H_2_**	**He**	**CO_2_/CH_4_**	**CO_2_/N_2_**	**O_2_/N_2_**	**He/CH_4_**
PEEK-WC [13] *	0.24	1.24	6.04	0.25	13.4	12.5	24.2	25.2	5.17	50.0
PEEK-WC/*m*-xy	0.07	0.68	3.41	0.10	7.75	8.58	34.1	48.7	9.71	85.8
PEEK-WC/Fura	2.25	10.1	52.7	4.27	40.1	30.9	12.3	23.4	4.49	7.24
**Membrane**	**Diffusivity [10^−12^ m^2^ s^−1^]**						** *α* ** **(*D_a_*/*D_b_*)**			
	**N_2_**	**O_2_**	**CO_2_**	**CH_4_**	**H_2_**	**He**	**CO_2_/CH_4_**	**CO_2_/N_2_**	**O_2_/N_2_**	**He/CH_4_**
PEEK-WC [13] *	0.45	2.02	0.58	0.14	135	529	4.14	1.29	4.49	3779
PEEK-WC/*m*-xy	1.02	3.86	0.85	0.20	188	708	4.25	0.83	3.78	3540
PEEK-WC/Fura	28.0	53.0	15.1	10.5	455	776	1.44	0.54	1.89	73.9
**Membrane**	**Solubility [cm^3^_STP_ cm^−3^ bar^−1^]**						** *α* ** **(*S_a_*/*S_b_*)**			
	**N_2_**	**O_2_**	**CO_2_**	**CH_4_**	**H_2_**	**He**	**CO_2_/CH_4_**	**CO_2_/N_2_**	**O_2_/N_2_**	**He/CH_4_**
PEEK-WC [13] *	0.39	0.46	7.77	1.33	0.07	0.02	5.84	19.9	1.18	0.015
PEEK-WC/*m*-xy	0.05	0.13	3.02	0.38	0.03	0.009	7.95	60.4	2.60	0.024
PEEK-WC/Fura	0.06	0.14	2.62	0.30	0.07	0.03	8.73	43.7	2.33	0.100

* literature data.

## Data Availability

Original data are available from the authors upon request.

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
