# Peer review of "PEEK–WC-Based Mixed Matrix Membranes Containing Polyimine Cages for Gas Separation"

_molecules, 2021, doi:10.3390/molecules26185557_

Round 1
Reviewer 1 Report
This manuscript introduces the gas separation performance of PEEK-WC/polyimine cage MMMs aiming to fill the gap of knowledge for this series of membrane. The manuscript highlighted the potential of fura and m-xy based polymine cages in enhancing gas permeability and selectivity of PEEK-WC, respectively. In general, this work is significant, novel and suitable for this journal. However, there are some aspects in the manuscript that need to be revised carefully.
Some important comments are listed below. Please refer to the attached pdf for other comments:
- The membrane preparation method (Section 3.3 and/or 2.3) is too simplified, especially no detail information about the PDMS coating described. The way/amount PDMS coated may affect the overall performance of the membrane. Was PDMS coated a full dense layer or just sealing the pinholes/cracks? How did the PDMS (layer) affect the membrane thickness (this might consequently affect the calculation of gas permeability)? How did authors quantify/justify that the PDMS (layer) didn’t affect the overall membrane performance (permeability and selectivity)? Pressure of gas permeation test is also missing.
- It’s unclear the source of solubility data presented in Table 1?
- The performance of PEEK-WC/polyimine MMMs is quite competitive with other commercial membranes but authors didn’t make much comparison. It's suggestive to add into the Figure 7 some literature data of other popular glassy membranes and PEEK MMMs (if available). The data of common glassy membranes can be found in Robeson's upper bound papers.
- Several writing errors are found in this version of manuscript such as:
+ Missing description for parameters in Eq 6 & 7
+ Cross-reference of Fig. 2 and Fig. S13 are incorrect. The mix of cross-reference for figures in manuscript and in supplementary document in a sentence creates some difficulty for reader to follow and refer to.
+ Please ensure acronym is introduced only at the very first time the term used.
+ Please recheck the grammar and punctuation, particularly errors of comma “,” and full stop “.” (refer to the pdf file for more details)

Author Response
Comments and Suggestions for Authors
This manuscript introduces the gas separation performance of PEEK-WC/polyimine cage MMMs aiming to fill the gap of knowledge for this series of membrane. The manuscript highlighted the potential of fura and m-xy based polymine cages in enhancing gas permeability and selectivity of PEEK-WC, respectively. In general, this work is significant, novel and suitable for this journal. However, there are some aspects in the manuscript that need to be revised carefully.
Answer:
We thank the reviewer for the critical and constructive comments. We understand the reviewer’s concerns and have made modifications accordingly. Detailed answers are given below.
Some important comments are listed below. Please refer to the attached pdf for other comments:
The membrane preparation method (Section 3.3 and/or 2.3) is too simplified, especially no detail information about the PDMS coating described.
Answer: A description of the PDMS coating procedure is now included in the revised manuscript:
“The two-component PDMS was used (ELASTOSIL® M 4601 A and B), and the components A and B were mixed in weight ratio 9:1 according to the instructions of the supplier without the use of a solvent. A film of ca. 25 µm of the resin was applied on the surface of the membrane by a casting knife. The coated membranes were kept at room temperature to complete the crosslinking in 24 h.”
The way/amount PDMS coated may affect the overall performance of the membrane. Was PDMS coated a full dense layer or just sealing the pinholes/cracks?
PDMS was coated as a full dense layer from the pure resin to avoid any possible swelling of either one of the two polymers in the membrane by the solvent for PDMS. The final coating layer was ≈ 25 µm thick.
According to the Henis and Tripodi model [1], the resistance of the membrane to gas permeation can be expressed as the sum of the contribution of the mixed matrix membrane (MMMs) and the coating layer of the PDMS (resistances-in-series model). In this case, considering that the permeability of for instance CO2 in the PDMS is two to three orders of magnitude higher than that in the MMM (≈ 3000 barrer for PDMS vs. ≈ 3-50 barrer for the MMM) the resistance of the PDMS layer to the permeation is negligible and is not expected to affect the transport properties, except that it is blocking the Knudsen diffusion through the pinholes (main principle of the defect healing).
[1] J.M.S. Henis, M.K. Tripodi, Composite hollow fiber membranes for gas separation: the resistance model approach, J. Memb. Sci. 8 (1981) 233–246. doi:10.1016/S0376-7388(00)82312-1.
How did the PDMS (layer) affect the membrane thickness (this might consequently affect the calculation of gas permeability)? How did authors quantify/justify that the PDMS (layer) didn’t affect the overall membrane performance (permeability and selectivity)?
Answer:
The thickness of the PDMS layer is quite high, and for some membranes in the same order of magnitude of the MMMs. Therefore, the thickness of the uncoated membranes was used for all calculations of the gas permeability. As explained in the previous point, the resistance to permeation by the PDMS layer is negligible, but the thickness is not. Therefore, the PDMS layer thickness should not be included in the calculation of the permeability coefficient. The selectivity is also not affected by the PDMS layer because it is dominated by the permeabilities of the MMM.
Pressure of gas permeation test is also missing.
Answer:
As already indicated in the experimental section, the pressure of the permeation tests was 1 bar, which is now also reported in the caption of figure 6
- It’s unclear the source of solubility data presented in Table 1?
Answer: The solubility data reported in Table 1 were obtained experimentally. The methodology is now reported in the text in the “Materials and Methods” section. As suggested by the reviewer in the pdf file, a footnote was added under Table 1 to specify the “X” meaning (i.e. insoluble).
- The performance of PEEK-WC/polyimine MMMs is quite competitive with other commercial membranes but authors didn’t make much comparison. It's suggestive to add into the Figure 7 some literature data of other popular glassy membranes and PEEK MMMs (if available). The data of common glassy membranes can be found in Robeson's upper bound papers.
Answer: Figure 7 was updated with data from Poly(aryl ether) membranes and Poly(aryl ether ketone) membranes, reported in the database of the Membrane Society of Australasia, available online.
- Several writing errors are found in this version of manuscript such as:
a+ Missing description for parameters in Eq 6 & 7
Answer: An additional equation (now Eq. 6) was inserted and all symbols of this and the other equations were explained in the text.
b+ Cross-reference of Fig. 2 and Fig. S13 are incorrect. The mix of cross-reference for figures in manuscript and in supplementary document in a sentence creates some difficulty for reader to follow and refer to.
Answer: As suggested by the reviewer in the pdf file, all X-Ray spectra are now moved to the main text, the graphs are numbered as a,b,c and d, as in other figures, and the text is rewritten as: “Fura and m-xy resulted to be the most crystalline samples among the investigated systems, as indicated by the reported XRPD patterns (Figure 2a,b). Instead, an amorphous halo is evident in both p-xy and the biphenyl cage (Figure 2c,d), where few peaks are superimposed on a very broad signal in the 10°-30° angular range.
c+ Please ensure acronym is introduced only at the very first time the term used.
Answer: We believe that all acronyms should be explained the first time they appear in the main text, even if few of them may have appeared in the abstract. Nevertheless, to avoid the problem, only the full names were used in the abstract. Only PEEK-WC was mentioned, being a name rather than an abbreviation.
d+ Please recheck the grammar and punctuation, particularly errors of comma “,” and full stop “.” (refer to the pdf file for more details). peer-review-13794180.v1.pdf
Answer: All minor corrections were made directly in the revised manuscript. The more important corrections are specified below. Answers are also inserted in the pdf file:
Line 89-92 (original). The punctuation was corrected (point instead of semicolon) and the text was slightly rewritten.
The original text “In the first step, the penetrating gas molecules dissolve on the feed side of the membrane; in the second step the diffusion of the gas molecules occurs through polymeric matrix and finally, in the third step, there is desorption from the opposite side of the membrane (permeate side)”
Is replaced by
“In the first step, the penetrating gas molecules dissolve in the membrane on the feed side. In the second step, they diffuse through polymeric matrix from the feed side to the opposite side (permeate). In the third and final step, the gas molecules desorb from the permeate side of the membrane into the permeate reservoir.”
Lines 135-138 (original): See answer 4b above
Lines 170-172 (original). The reviewer questions the meaning of ‘visually homogeneous’.
Homogeneous refers to the macroscopic scale and in the case of mixed.matrix membranes it is not uncommon that the films show visibly different regions with different colour or opacity due to a poor or uneven dispersion of the filler materials. In our case, the samples are only somewhat opaque due to the different refractive index of the matrix and the filler, but they do not show clearly visible inhomogeneities.
The text is rewritten as: “Thus, the successfully prepared MMMs PEEK-WC/m-xy and PEEK-WC/Fura did not show any macroscopic defects or inhomogeneities visible to the naked eye, suggesting an even dispersion of the fillers. They were only slightly opaque due to the different refractive index of the polymer and the filler materials and had a thickness 42 μm and 78 μm for PEEK-WC/m-xy and PEEK-WC/Fura, respectively,
Figure 5: the scale bars are now indicated in the first set of images.
Line 195-197 (original): the position and the slope of the baseline refer to the start of the time lag curve. The full equation of the curve (as a Tailor series) is now added as the new Eq. 6 in which the first two terms refer to the starting pressure and the baseline slope, respectively. The text of section 2.3 is slightly modified and rephrased as:
“An example of two permeation curves, determined by the so-called time lag method in a fixed-volume pressure-increase setup, and described by Eq. 6, is given in Figure 6. The permeate pressure of CO2 in PEEK-WC/m-xy membrane is plotted as a function of time, before and after defect healing with PDMS. From the immediate pressure increase and the very steep slope of the uncoated sample, it is evident that the PDMS coating is needed to correct pinhole defects and to obtain a curve where the determination of the time lag is clear and well defined. This allows the determination of the gas transport parameters of the MMM, i.e. the permeability and diffusion coefficients of the gases and, indirectly, the solubility. The flat baseline in the PDMS-coated membrane, i.e. the tangent to the very initial part of the curve defined by the term (dp/dt)0 in Eq. 6 and Eq. 7, confirms that leak flow through remaining pinhole defects is negligible for CO2. Wherever this is not the case, a baseline correction was applied via (Eq. 7). As described previously, this procedure allows the correct calculation of the values of P, D and S of membranes with few defects [38].”
and in the experimental part above (Eq. 6):
“The time lag method was used for the determination of the permeability (P), diffusion (D) and solubility coefficients (S), which can be obtained from the increase of the permeate pressure, pt, as a function of time, t, after exposure of the membrane to the gas [49]:
where p0 and (dp/dt)0 are the starting pressure and baseline slope, respectively, which should be negligible in a well-evacuated and leak free membrane and permeability instrument. R is the universal gas constant, T the absolute temperature, VP the permeate volume, Vm the molar volume of a gas at standard temperature and pressure [22.41·10‑3 m3STP mol‑1 at 0°C and 1 atm] , A the exposed membrane area, l its thickness, pf the feed pressure, S the gas solubility and D the diffusion coefficient.
The permeability P was obtained from the permeation curve (Eq. 7) in the pseudo steady-state:”

Reviewer 2 Report
The paper with the tittle of "PEEK-WC-based Mixed Matrix Membranes Containing Polyi-2 mine Cages for Gas Separation" is about study of the gas transport properties in MMM membranes. The paper is well written and the methods were described well. However I have a concern about permeability data and results from PEEK-WC/Fura membrane since this film was defective and coated with PDMS.
the two organic cages were selected to make MMM membranes are m-xy and Fura. The membrane with Fura was defective and thus PDMS coating was used to reduce the effect of the defects. Generally additives to the polymer films may increase selectivity of the composite films. Lower He/CH4 and CO2/CH4 in PEEK-WC/Fura film than pristine PEEK-WC film may suggests that the contribution of coated defects in gas permeabilities maybe much higher than MMM membrane itself.
If the numbers of defects are limited the the gas permeability through MMM membrane is dominant, however in highly defective membrane coated with PDMS the gas permeability result may not represents the MMM membrane but may show the permeability through PDMS layer as well. We need to know thickness, gas permeability and selectivity through PDMS Elastosil M 4601 layer to evaluate if the permeability and selectivity data is about PDMS layer or MMM membrane.
knowing the permeability and thickness of PDMS layer authors should use a model to identify what fraction of gas permeability in the membrane is about coated defective areas and what fraction is passing through PEEK-WC/Fura film.
Author Response
Comments and Suggestions for Authors
The paper with the tittle of "PEEK-WC-based Mixed Matrix Membranes Containing Polyimine Cages for Gas Separation" is about study of the gas transport properties in MMM membranes. The paper is well written and the methods were described well. However I have a concern about permeability data and results from PEEK-WC/Fura membrane since this film was defective and coated with PDMS.
Answer:
We thank the reviewer for the critical and constructive comments. We understand the reviewer’s concerns and have made modifications accordingly. Detailed answers are given below.
the two organic cages were selected to make MMM membranes are m-xy and Fura. The membrane with Fura was defective and thus PDMS coating was used to reduce the effect of the defects. Generally additives to the polymer films may increase selectivity of the composite films. Lower He/CH4 and CO2/CH4 in PEEK-WC/Fura film than pristine PEEK-WC film may suggests that the contribution of coated defects in gas permeabilities maybe much higher than MMM membrane itself.
Answer: See also the answer to Reviewer#1, point 1:
Indeed, the defects dominate the transport until they are covered, while after PDMS coating the transport is governed by the solution-diffusion mechanism though the MMM material. Besides blocking the Knudsen diffusion through the pinhole defects (the principle of the defect healing procedure), the effect of PDMS on the resistance to the transport through the membrane (and thus on its permeability and selectivity) is negligible. There is no reason to believe that PDMS could be compatible with the PEEK-WC and affect its properties by plasticization or blend formation. Instead, solvents are more likely to interact with the membrane or the cages, and therefore a solvent-free coating method was used. For more detailed discussion we refer to the answer to Reviewer #1.
If the numbers of defects are limited the the gas permeability through MMM membrane is dominant, however in highly defective membrane coated with PDMS the gas permeability result may not represents the MMM membrane but may show the permeability through PDMS layer as well. We need to know thickness, gas permeability and selectivity through PDMS Elastosil M 4601 layer to evaluate if the permeability and selectivity data is about PDMS layer or MMM membrane.
Answer: (see also point 1 of Reviewer #1)
Coating with PDMS for the correction of pinholes in membranes for gas separation is widely used ([2–5]) and its effect on the diffusive transport through the dense membrane is negligible because there are at least several orders of magnitude difference in their intrinsic permeability coefficients. Instead, it completely blocks the Knudsen diffusion through the pinhole defects, which, in turn, is several orders of magnitude faster than the diffusive transport even through PDMS (Figure 6). Unless a very high percentage of the surface is defective, the properties of PDMS might start dominating, but in that case the selectivity of the membrane should approach that of PDMS. This is NOT the case, and therefore we can safely conclude that the only effect of PDMS is to block Knudsen diffusion and to restore the intrinsic properties of the MMM.
PDMS was coated as a full dense layer from the pure resin to avoid any possible swelling of either one of the two polymers in the membrane by the solvent for PDMS. The final coating layer was ≈ 25 µm.
According to the Henis and Tripodi model [1], the resistance of the membrane to gas permeation can be expressed as the sum of the contribution of the mixed matrix membrane (MMMs) and the coating layer of the PDMS (resistances-in-series model). In this case, considering that the permeability of for instance CO2 in the PDMS is two to three orders of magnitude higher than that in the MMM (≈ 3000 barrer for PDMS vs. ≈ 3-50 barrer for the MMM) the resistance of the PDMS layer to the permeation is negligible and is not expected to affect the transport properties, except that it is indeed blocking the Knudsen diffusion through the pinholes.
[1] J.M.S. Henis, M.K. Tripodi, Composite hollow fiber membranes for gas separation: the resistance model approach, J. Memb. Sci. 8 (1981) 233–246. doi:10.1016/S0376-7388(00)82312-1.
[2] M.S. Suleman, K.K. Lau, Y.F. Yeong, Characterization and Performance Evaluation of PDMS/PSF Membrane for CO2/CH4 Separation under the Effect of Swelling, in: Procedia Eng., 2016. doi:10.1016/j.proeng.2016.06.525.
[3] B. Haider, M.R. Dilshad, M. Atiq Ur Rehman, J. Vargas Schmitz, M. Kaspereit, Highly permeable novel PDMS coated asymmetric polyethersulfone membranes loaded with SAPO-34 zeoilte for carbon dioxide separation, Sep. Purif. Technol. (2020). doi:10.1016/j.seppur.2020.116899.
[4] S.S. Madaeni, M.M.S. Badieh, V. Vatanpour, N. Ghaemi, Effect of titanium dioxide nanoparticles on polydimethylsiloxane/ polyethersulfone composite membranes for gas separation, Polym. Eng. Sci. (2012). doi:10.1002/pen.23223.
[5] S.S. Madaeni, M.M.S. Badieh, V. Vatanpour, Effect of coating method on gas separation by PDMS/PES membrane, Polym. Eng. Sci. (2013). doi:10.1002/pen.23456.
A comment has been added to the section 2.2
knowing the permeability and thickness of PDMS layer authors should use a model to identify what fraction of gas permeability in the membrane is about coated defective areas and what fraction is passing through PEEK-WC/Fura film.
Answer: Indeed, we understand the concern of the reviewer about the possible effect of PDMS coating. However, this concern is only justified for membrane materials with a permeability close to that of PDMS (for which a thick PDMS layer would have a non-negligible resistance to transport according to the resistances-in-series model [1]), or membranes in which a high percentage of the surface is represented by pores (for which a non-negligible part of the transport would take place through the PDMS-filled pores according to the parallel resistances model [1]). However, our membranes are very far away from both situations, and in this case, the only effect of PDMS is to seal the defects and suppress the Knudsen diffusion.
Any attempt to quantify the contribution of the resistance of the PDMS on top of the MMM to the overall resistance, or the contribution of gas flux through the PDMS in the pinholes compared to the overall flux, would not exceed the experimental error in for instance the measurement of the membrane thickness or the effective membrane area. It would most likely also distract the attention of the reader from the actual results, and therefore we believe that it is sufficient to state that the contribution of PDMS is negligible.